# Breath Volatile Organic Compounds in Surveillance of Gastric Cancer Patients following Radical Surgical Management

**DOI:** 10.3390/diagnostics13101670

**Published:** 2023-05-09

**Authors:** Roberts Škapars, Evita Gašenko, Yoav Y. Broza, Armands Sīviņš, Inese Poļaka, Inga Bogdanova, Andrejs Pčolkins, Viktors Veliks, Valdis Folkmanis, Anna Lesčinska, Inta Liepniece-Karele, Hossam Haick, Ingrīda Rumba-Rozenfelde, Mārcis Leja

**Affiliations:** 1Institute of Clinical and Preventive Medicine, Faculty of Medicine, University of Latvia, LV-1586 Riga, Latvia; 2Department of Abdominal and Soft Tissue Surgery, Oncology Center of Latvia, Riga East University Hospital, LV-1038 Riga, Latvia; 3Department of Chemical Engineering and Russel Berrie Nanotechnology Institute, Technion—Israel Institute of Technology, Haifa 3200003, Israel

**Keywords:** volatile organic compounds, gastric cancer, nanosensors, cancer follow-up, gas chromatography-mass spectrometry

## Abstract

As of today, there is a lack of a perfect non-invasive test for the surveillance of patients for potential relapse following curative treatment. Breath volatile organic compounds (VOCs) have been demonstrated to be an accurate diagnostic tool for gastric cancer (GC) detection; here, we aimed to prove the yield of the markers in surveillance, i.e., following curative surgical management. Patients were sampled in regular intervals before and within 3 years following curative surgery for GC; gas chromatography-mass spectrometry (GC-MS) and nanosensor technologies were used for the VOC assessment. GC-MS measurements revealed a single VOC (14b-Pregnane) that significantly decreased at 12 months, and three VOCs (Isochiapin B, Dotriacontane, Threitol, 2-O-octyl-) that decreased at 18 months following surgery. The nanomaterial-based sensors S9 and S14 revealed changes in the breath VOC content 9 months after surgery. Our study results confirm the cancer origin of the particular VOCs, as well as suggest the value of breath VOC testing for cancer patient surveillance, either during the treatment phase or thereafter, for potential relapse.

## 1. Introduction

GC is the fourth most common cause of cancer-related death in the world [1], with an overall 5-year relative survival rate of about 20% in most areas of the world, except in Japan, where 5-year survival rates are above 70% because most GC cases are found in stages I and II [2]. Such high survival rates may be due to the effectiveness of mass screening programs in Japan [3]. It is projected that the annual incidence of GC will rise significantly in the coming years, with an estimated 1.8 million new cases and 1.3 million deaths expected by the year 2040 [4].

The primary treatment choice for locoregional GC remains surgery [5]. Even in combination with neoadjuvant or adjuvant chemotherapy, the overall survival rate stays low for node-positive disease [6]. It is noteworthy that most instances of GC recurrence happen within the first two years after local therapy, accounting for approximately 70% to 80% of cases. Moreover, almost all relapses occur within five years after curative treatment, which accounts for around 90% of cases. The postoperative surveillance strategies for patients who undergo curative resection for GC are still a matter of debate [7]. The guidelines for surveillance strategies after curative intent resection for gastric cancer proposed by various scientific societies and groups—such as the NCCN, the ESMO, the Association of Upper Gastrointestinal Surgeons of Great Britain and Ireland, the British Society of Gastroenterology and the British Association of Surgical Oncology, the Italian Research Group for Gastric Cancer, and French guidelines—exhibit significant discrepancies. This heterogeneity of the guidelines can be attributed to the absence of robust evidence to support them [8].

The standard diagnostic tools for GC follow-up after treatment are upper endoscopy, computed tomography, and the tumor markers (carcinoembryonic antigen (CEA), cancer antigen 19-9 (CA 19-9), and carbohydrate antigen 72-4 (CA 72-4)) [8]. Routine biomarkers or imaging methods often have limited sensitivity in detecting early microscopic metastases. Routine biomarkers or imaging poorly detect early microscopic metastases. CEA and CA 19-9 are the most-used biomarkers for follow-up; however, they detect only 40% of cancer recurrence [9]. Thoracoabdominal computer tomography is able to detect more than 90% recurrences, while physical exam and upper endoscopy near 10% recurrences [9]. According to European Society of Medical Oncology guidelines and National Comprehensive Cancer Network guidelines, computed tomography is recommended from 6 weeks up to 12 month intervals depending on the stage and therapy received [7,10]. Nevertheless, only 26% of patients with recurrence were able to receive therapy [9]. Positron emission tomography (PET), combined with computed tomography (CT)-PET-CT has been reported to exhibit a sensitivity of 91.2%, and a specificity of 61.5%, in detecting post-surgical recurrence of GC. The NCCN guidelines on the use of FDG PET-CT in gastric carcinoma have been fairly permissive, despite the lack of robust evidence. The guidelines recommend the use of FDG PET-CT for surveillance following neoadjuvant and adjuvant therapy in stage I–III disease [11].

It is therefore important to identify biomarkers that could predict the ensuing development of metastatic disease. Such biomarkers could predict high-risk patients where additional or more aggressive adjuvant therapy could be optioned.

Breath analysis is a young and promising field of research with its roots back in 460–370 years before Christ, when Hippocrates described *fetor oris* and *fetor hepatis.* Since then, sensitivity and speed of breath analysis has greatly improved. This direction of science has made analysis of VOCs from exhaled breath feasible.

Human breath contains more than 10,000 different VOCs in very low concentrations measured in parts-per-billion to parts-per-trillion. VOCs can be the endogenous origin or exogenous origin [12]. Endogenous VOCs originate in the human body as a result of normal metabolism or a result of pathologic disorders. Exogenous VOCs can penetrate the human organism from environmental exposure, and can accumulate in body tissues for many years. There are many methods to diagnose different VOCs in exhaled breath. The most important method to be noted is GC-MS. This method identifies and quantifies various VOCs exhaled. Another method is nanosensor technologies. This method deploys cross-reactive nanoarrays in a combination of pattern recognition methods. This approach provides collective VOC patterns rather than identification and quantification of specific VOCs.

Detecting and measuring breath VOCs become a very promising approach for the early detection and screening of various cancers, as well as other diseases [13,14,15,16]. Our research and studies from other research groups across the globe have concentrated on the potential role of VOCs for GC detection [17,18,19,20,21,22,23]. Despite limited evidence, 23.4% of specialists in the field approved the use of breath-tests in GC screening, as reported in a previously conducted survey [24].

After GC treatment, routine screening for asymptomatic recurrence is recommended for a 5-year period by a group of international experts [7,10,25]. As of today, there is a lack of a perfect non-invasive test for the surveillance of patients for potential relapse following curative treatment. If a marker is related to cancer, one would expect that following a radical cancer treatment—either radical surgery, radiation, or medical therapy—the signs of the marker would decrease or disappear. This is how traditional gastric-related cancer biomarkers (CEA, CA 19-9, CA 72-4) react after cancer treatment.

Therefore, within the current study, we aimed to identify whether the potentially GC-related VOC markers get changed following curative cancer surgery, and whether this change may be observed in markers being detected with either GC-MS or nanosensor approaches.

## 2. Materials and Methods

### 2.1. Study Population

The study was run from October 2012 until December 2021 (including the surveillance period); altogether, 51 patients with gastric adenocarcinoma naïve for previous specific therapy were recruited at the Department of Abdominal and Soft Tissue Surgery, Oncology Centre of Latvia, Riga East University Hospital before planned radical GC surgery. Four patients were excluded because distant metastasis were revealed at the time of recruitment. All patients underwent curative surgical procedures performed by one surgical unit. All patients underwent radical distal or total gastrectomy with D2 lymph node dissection, as defined by the Japanese Gastric Cancer Association [26]. Therefore, 47 patients were available for the follow-up.

For the purpose of the current analysis, we did not include the patients that were developing the metastatic disease (N.12) or secondary cancer (N.3), nor those with surgical complications (N12). Finally, 20 patients had completed the 5-year follow up and those data were analyzed (see Figure 1 for the study Flow-chart). Chemotherapy or radiation therapy during the observational period was another indication for exclusion from the final analysis.

### 2.2. Sampling Protocol

Sampling was performed 1–3 days before surgical treatment (T0), followed by scheduled sampling every three months during the first year (at least 6 months), and every 6 months from 2nd year of follow-up until the 36th month after surgery (see Figure 1). From the 3rd month after surgery, patients were followed by one oncologic surgeon, and diagnostic procedures were done as recommended in the National Comprehensive Cancer Network Clinical Practice Guidelines in Oncology [7]. Each time a patient was attending the physician for follow-up, breath tests were performed.

### 2.3. Breath Collection

Exhaled breath samples were collected precisely as described by Amal et al. [17]. Detailed information is available in Appendix A. Briefly, room air impurity was purified from the inhaled air by a lung washout, 3 min of inhalation through a mouthpiece with a filter cartridge on the inspiratory port mouthpiece to diminish the exogenous VOCs concentration. Two samples from each patient were collected into a 1-litre Tedlar^®^ PVF collecting bag (Cel Scientific corporation, Cerritos, CA, USA). The VOCs in the breath samples were trapped and pre-concentrated in two-bed ORBO^TM^ 420 Tenax^®^ TA sorption tubes (purchased from Sigma-Aldrich, Beijing, China), and kept under refrigeration at 4 °C until they were transferred (under refrigeration) to the laboratory amenities for the breath analysis (Laboratory for Nanomaterial-Based Devices, Technion, Haifa, Israel).

Two different methods were applied. The first method we used was GC-MS to identify and quantify different VOCs in each comparison group. Breath VOCs were analyzed using a GC-MS-QP2010 instrument with an SLB-5ms capillary column and thermal desorption (TD) system. Samples were transferred from ORBOTM 420 Tenax^®^ TA sorption tubes to empty glass TD tubes, heat treated at 270 °C, and injected into the GC-system in splitless mode. The compounds were identified using spectral library match NISTL.14, and external standards were used to confirm the identity and quantity of the compounds. Gaseous standards were derived using a permeation/diffusion tube dilution system, and VOC concentration was determined by controlling the mass flow rate of the vaporized VOC(s). Calibration samples were analyzed under the same experimental conditions as the breath samples (detailed information can be found in the Appendix A).

The second method deployed cross-reactive nanoarrays in combination with pattern recognition methods. The study analyzed breath samples using a sensor array consisting of eight sensors based on two types of nanomaterial: organically stabilized spherical gold nanoparticles (GNPs) and single-walled carbon nanotubes (SWCNTs). The organic ligands of the GNPs provided broadly cross-reactive absorption sites for the breath VOCs. A Keithley data logger device was used to acquire resistance readings from the sensor array during the experiment. Four sensing features were extracted from the time-dependent resistance response of each sensor that related to the normalized resistance change at the beginning, middle, and end of the exposure, and to the area beneath the time-dependent resistance response. Discriminant Factor Analysis (DFA) was used to obtain breath patterns from the collective response of the sensors. DFA is a linear, supervised pattern recognition method that reduces the multidimensional experimental data, in which the classes to be discriminated are defined before the analysis is performed. DFA was also used to select the sensors with the most relevant organic functionality out of the repertoire of twenty-one by filtering out non-contributing sensors. It is important to note that nineteen of the forty sensors were excluded before analysis because of their “noise” response to the breath samples. The study found that each sensor responded to all (or to a certain subset) of the VOCs found in the exhaled breath samples. This approach provides collective VOC patterns rather than the identification and quantification of specific VOCs (detailed information can be found in the Appendix A).

Contaminants from the sample room were obtained and taken in the notice, see Appendix A.

### 2.4. Chemical Analysis

The GC-MS chromatograms were analyzed using Mass Hunter qualitative analysis (version B.07.00) followed by Mass Hunter quantitative analysis (version B.07.01; Agilent Technologies, Santa Clara, CA, USA). The compounds were tentatively identified through spectral library match NISTL.14 (accessed on June 2014, National Institute of Standards and Technology, Gaithersburg, MD, USA). The Kruskal-Wallis test was used to compare samples before the operation, and scheduled samples after the operation.

The comparison was made between samples before GC curative surgery, and samples after surgery in a specified order (see Figure 2). We used the Kruskal-Wallis test to determine at what period breath prints change after the operation. For this purpose, we developed 8 paired Discriminant Factor Analysis (DFA) models for comparison. Each paired model included preoperative (T0) samples and, accordingly, follow-up samples from 3 months (T1) after the operation until 36 months (T8) after the operation (see Figure 2).

### 2.5. Statistical Analysis

In the GC-MS analysis, VOCs showing significant differences (cut-off values: 0.05) between pre-surgery and post-surgery GC were determined using the Kruskal-Wallis test, a nonparametric version of the classical one-way ANOVA for populations whose data cannot be assumed to be normally distributed. In the nanosensor group, the paired pre-surgery and post-surgery responses were calculated using the Area Under Curve method, performing the following statistical procedures: analysis of variance ANOVA to determine cancer remission gradations, time of taken breath samples, and the nonparametric Kruskal-Wallis test. Breath sample comparisons for both methods were analyzed with MATLAB R2020a (MathWorks Inc., Natick, MA, USA)

## 3. Results

The study compared nanosensors and GC-MS for the detection of VOCs in breath samples from patients undergoing surgery. Table 1 displays the number of useful samples obtained for each group at various time points, along with the total number of patients involved in the study. Overall, 198 samples were collected from 20 patients, 2 samples at each scheduled time (Table 1). Similar numbers of useful samples were collected across time points and methods. The number of patients involved in the study decreased over time as some patients dropped out of the study or were lost to follow-up. In first 12 months of follow-up, 9 (45%) of the patients discontinued participation in the study.

The study group consisted of a total of 20 participants, out of which 16 were men and 4 were women. The gender distribution in the study group was predominantly male, with 80% of the participants being men. The median age of the participants was 65 years, with a range of 40 to 81 years. All the participants in the study group were of Caucasian origin.

The stage of the disease was assessed as I and II in the majority of the participants (15 out of 20), and the remaining 5 participants had III disease. The smoking status of the participants was also recorded as part of the study. Of the 20 participants, 15 were either current or ex-smokers, while the remaining 5 were non-smokers. Table 2 provides a detailed breakdown of the smoking status of the study group.

According to the Lauren classification, the frequency of intestinal cancer was observed to be 5 out of 20 (25%), and diffuse cancer was 6 (30%). The majority of the participants 9 (45%) had mixed-type gastric adenocarcinomas.

The differentiation grade of the cancerous tissues was evaluated for all 20 participants. Out of these, 1 participant had well-differentiated adenocarcinoma, while the majority of participants (11 out of 20) had moderately differentiated adenocarcinomas. Additionally, 7 participants were diagnosed with poorly differentiated and one participant was found to have undifferentiated adenocarcinoma. By study design, all participants underwent surgical treatment, with 14 patients receiving total gastrectomy and 6 patients undergoing distal gastrectomy. Table 2 displays the full data for the variables.

### 3.1. Chemical Analysis

The chemical composition of breath samples from 20 patients with GC was compared before and repeatedly after surgery (see Figure 2). Altogether, 100 GC-MS samples were analyzed. Five samples were excluded because of technical problems. As a result, 19 GC-MS preoperative samples were compared with samples after operation in a precise manner as described in the Methods and Appendix A.

Altogether, 152 chemicals were found in at least 90% of samples. All chemicals had main mass in the range 41–119.1 *m*/*z*, and retention times of 2.18–39.71 min.

The GC-MS analysis revealed one VOC (14b-Pregnane) that significantly reduced twelve months after the operation, and three VOCs (Isochiapin B, Dotriacontane, Threitol, 2-*o*-octyl-) that were found to be decreased after 18 months.

### 3.2. Nanosensor Analysis

A total of 98 samples met inclusion criteria for analysis from the study group. Seven samples were excluded because of technical reasons.

All breath samples were analyzed using eight sensors based on two types of nanomaterial: (a) organically stabilized spherical gold nanoparticles (GNPs); and (b) single-walled carbon nanotubes (SWCNTs). Six different organic functionalities of the GNP sensors and two for the SWCNT contributed to the chemical diversity of the sensors.

The comparison between samples before GC curative surgery and samples after surgery (see Figure 2) determined that nanomaterial-based sensor numbers S9 and S14 (*p* < 0.05) confirmed breath changes 9 months after operation (see Figure 3).

## 4. Discussion

GC ranks as one of the leading causes of cancer-associated death worldwide [1]. Developing cancer metastasis plays a crucial role in patient survival. Today, the metastatic disease after GC surgery is diagnosed because of developed symptoms and/or positive imaging results. There is a crucial need for a marker that could diagnose or at least arouse suspicion that micrometastases are developing, a marker that would initiate or prolong adjuvant therapy. This marker would also suggest that such patients should be followed more closely after surgery.

An ideal cancer biomarker should be found altered at the time of cancer diagnosis (ideally—already in the preclinical phase), then demonstrate normalisation of the result in the case of curative treatment, followed by altered results once more in the case of cancer relapse [27,28]. Such biomarkers are lacking or are inaccurate in the case of many cancers, including GC [9,29]. Furthermore, if a biomarker fulfills these characteristics, this is likely to be originated by cancer, but not other confounding factors. The results from our own group have suggested the change in the breath VOC measurements in patients receiving treatment resulting in gut microbiota perturbation [30].

The breath VOC diagnostic principle has been demonstrated to be very attractive for many cancers, including GC [31,32,33]; however, there is very limited information published so far on the potential use of VOC detection for cancer surveillance. According to our knowledge, this is the first study having addressed the role of VOCs in GC. There are few studies available where the VOC concept was used following lung cancer surgery by suggesting that the concept could be applied not only for diagnostic but also for follow-up purposes [34,35,36]. Broza et al. [34] found 5 VOCs (2-Methyl-1-pentene, 2-Hexanone, 3-Heptanone, Styrene, 2,2,4-Trimethyl-hexane) decreasing after LC resection. In this study, nanosensor technology showed an accuracy of 80% for distinguishing the pre-surgery and post-surgery LC states. Poli et al. found that only two of five LC-specific VOCs (isoprene and decane) decreased in a short term period after LC surgery [35], and two VOCs (isoprene and benzene) decreased after 3 years after LC resection) [36]. The authors concluded that larger clinical studies are required to validate these findings.

A considerable number of previous studies of our group or others have suggested the role of VOCs in GC detection [17,18,19,20,21,22,23,37,38]; however, proof of the justification of the concept still remains vague.

In the current study, we identified 152 chemicals from breath samples. Among these chemicals, only one (isoprene) chemical was related to previously detected GC VOCs [18,19]. Isoprene did not decrease during the three years of the follow-up period in our study. However, we confirm another VOC decreased during the follow-up time. 14b-Pregnane statistically decreased twelve months after surgical treatment, and another three VOCs (Isochiapin B, Dotriacontane, Threitol, 2-*o*-octyl-) decreased 18 months after treatment (*p* = 0.05).

14b-Pregnane was found in *Geranium saxatile* [39]. 14b-Pregnane was also extracted from marine nature products such as soft corals [40]. Furthermore, 14b-Pregnane was extracted from *Sarcophyton* and *Sinularia* coral species, and were indicated as having a potential anti-inflammatory effect [41].

Additionally, isochiapin B was found present in *Commiphora myrrha* resin ethanolic extract widely used in cosmetics [42]. This extract is described as antiseptic, anti-fungal, and anti-inflammatory, and promotes wound healing. Isochiapin B is also found to be in bioactive compounds, furthermore, such natural compounds have anticarcinogenic properties [43]. Dotriacontane has been detected in natural products like coconut, papaya, and tea. It can also be a natural component of tobacco and cigarette smoke [44]. The exact origin of all the identified chemical substances that were found in our study is not known. Over 220 different cancer-related (lung, colorectal, breast, esophageal, gastric, head, and neck) VOCs were found in reviews [13,16]. Most of the reviewed studies are applicable to lung cancer. However, these studies still cannot define lung cancer-related VOCs.

Alongside the GC-MS method, we applied the sensor-based method. For this purpose, we developed 8 paired DFA models for comparison. We compared preoperative samples to follow-up samples in the described manner (see Section 2 and Figure 2). Two from various sensors showed different breath print pattern samples from the ninth month after surgery (see Figure 3). This could further support the hypothesis that nanosensors can react to cancer.

According to our knowledge, this was the first study to investigate the dynamics of markers in GC following radical management of GC, and one of a few studies addressing VOC dynamics within surveillance of any cancer type. Developing metastatic disease, unwillingness to continue with the surveillance, as well as technical limitations, have been the major factors that decrease the current study group from a significantly higher number of patients that were recruited initially. Nevertheless, we were able to complete the 3-year follow-up for a substantial number of patients.

In conclusion, this is the first study to suggest that VOC detection could be a valuable tool for GC patient surveillance following a curative surgical management. Based on the obtained results, we hypothesize that this approach could also be used in medically managed patients (in the situation of neoadjuvant and adjuvant treatment), and potentially applied in other cancer types. Large-scale implementation studies with a sufficiently long follow-up period are required.

## Figures and Tables

**Figure 1 diagnostics-13-01670-f001:**
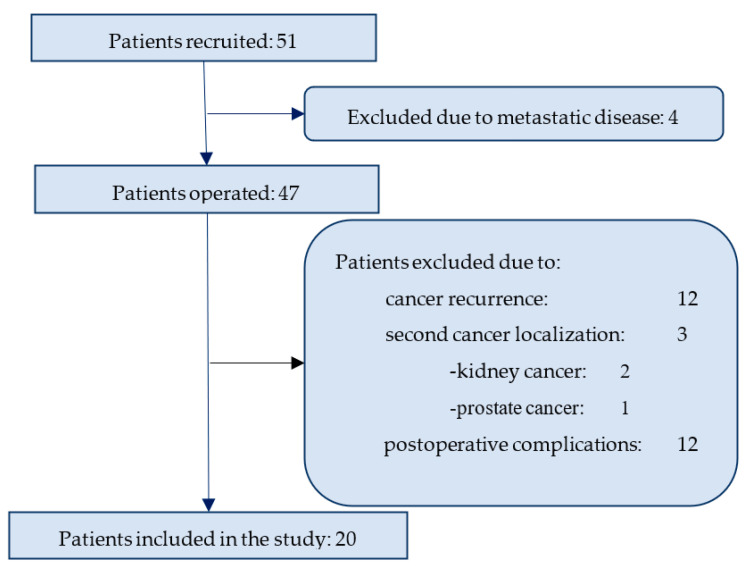
Flow chart for a patient exclusion process.

**Figure 2 diagnostics-13-01670-f002:**
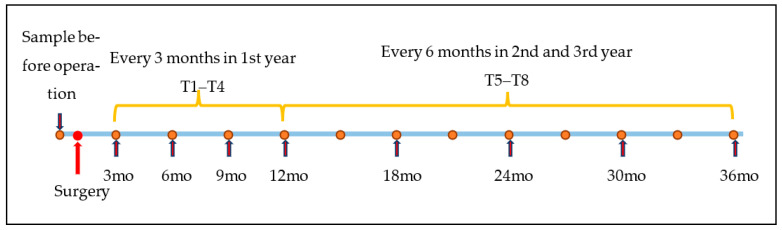
Description of sampling intervals. T—time, mo—a month after the operation. Blue arrows showing breath test time.

**Figure 3 diagnostics-13-01670-f003:**
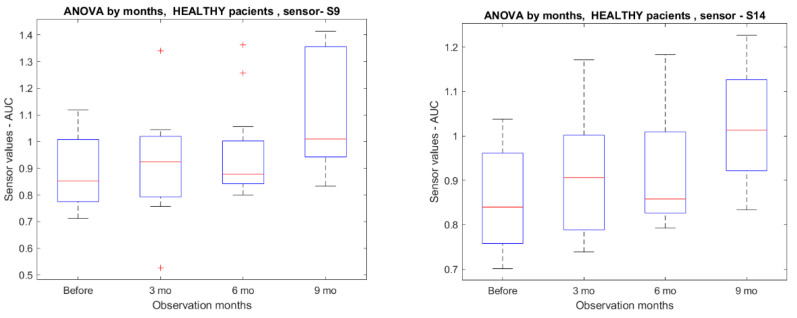
Showing different breath prints nine months after the curative operation. Mo—months.

**Table 1 diagnostics-13-01670-t001:** Useful sample count in each group.

	T0 Preop	T1 (3 mo)	T2 (6 mo)	T3 (9 mo)	T4 (12 mo)	T5 (18 mo)	T6 (24 mo)	T7 (30 mo)	T8 (36 mo)	Total
nanosensors	18	14	15	10	11	10	7	7	6	98
GC-MS	19	14	16	10	11	10	7	7	6	100
Patients involved	20	14	17	10	11	11	7	7	7	104

Preop—1–3 days before the operation, mo—months after the operation.

**Table 2 diagnostics-13-01670-t002:** Summary of personal data.

	Number or Mean (Standard Deviation)	%
**Total Number of Patients**	**20**	
Sex, *n*		
Male	16	80
Female	4	20
Mean age, y	61.5 ± 9.9	
Smoking, *n*		
Smoker	7	35
Ex-smoker	8	40
Non-smoker	5	25
Cancer stage, *n*		
Stage I	7	35
Stage II	8	40
Stage III	5	25
Cancer type, *n*		
Intestinal	5	25
Diffuse	6	30
Mixed	9	45
Cancer differentiation grade, *n*		
Well-differentiated	1	5
Moderately differentiated	11	55
Poorly differentiated	7	35
Undifferentiated	1	5
The operation type, *n*		
Total gastrectomy	14	70
Distal gastrectomy	6	30

*n*—number, y—years.

## Data Availability

Not applicable.

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
