# Peer review of "Breath Volatile Organic Compounds in Surveillance of Gastric Cancer Patients following Radical Surgical Management"

_diagnostics, 2023, doi:10.3390/diagnostics13101670_

Round 1

Reviewer 1 Report

This manuscript describes the utillization of breath volatile organic compounds as an accurate diagnostic tool forGC detectionin. The topic is of great interesting and importance, especially for the development non-invasive diagnostic tool. The obtained results by GC-MS and nanosensor proved the potantial of VOCs for GC detection. In general, the manuscript is well written and organized. However, some points need to be addressed before it is suitable for publication.

1. It is not clear how avoid the contamination from mouth during sample collection.

2. It is better to further emphasize why the GC-MS and nanosensor were used simultaneously for breath VOCs detection.

3. Why the resluts of GC-MS and nanosensor seems not comparbale?

4. The detection mechanism of nanosensor for VOCs detection is not clear. What about the sensitivity and specificity of nanosensor?

Author Response

Dear Reviewer 1,

Thank you for your constructive comments, which helped us to focus and improve the manuscript. Please see the changes in the attached letter that were made according to your remarks.

Kind regards,

Roberts Skapars,

on behalf of the authors.

Reviewer 2 Report

This manuscript is an original article that investigated the availability of breath volatile organic compounds (VOCs) in surveillance of gastric cancer patients following curative surgical management. The authors identified four VOCs decreased at 12-18 months following surgery and two nanomaterial-based sensors were changed after surgery.

Breath VOCs in screening and surveillance of cancer is a promising approach as it’s a simple, non-invasive manner.

This study was conducted well, and the methods are appropriate. The data are presented clearly.

The results will be of interest to researchers and clinicians in the field.

However, the following minor issues require clarification:

Minor

1.     (Introduction) Most of the introduction is occupied by the description of methods in surveillance of gastric cancer patients following curative surgical management. Please inform readers of already-known knowledge regarding breath VOCs in the field of cancer in more detail.

2.     The authors should describe limitations in this study.

3.     (P6L16-19) These sentences are duplicated.

4.     (P8L34) “Antiseptic” is duplicated.

Author Response

Dear Reviewer 2,

Thank you for your constructive comments, which helped us to focus and improve the manuscript. Please see the changes that were made according to your remarks.

We believe that the changes made have resulted in an improved revised manuscript, which is included alongside our response letter.

Kind regards,

Roberts Skapars,

on behalf of the authors.
